



**The effect of salinity, light regime and food source on C and N uptake in a kleptoplast-bearing**
**foraminifera**
**Michael Lintner[1], Bianca Lintner[1], Wolfgang Wanek[2], Nina Keul[3] and Petra Heinz[1]**
**[1]University of Vienna, Department of Palaeontology, Vienna, Austria**
**[2]University of Vienna, Department of Microbiology and Ecosystem Science, Division of Terrestrial Ecosystem**
**Research, Vienna, Austria**
**[3]Christian-Albrechts-Universität zu Kiel, Institute of Geosciences, Germany**
**Abstract**
Foraminifera are unicellular organisms that play an important role in marine organic matter cycles. Some species are
able to isolate chloroplasts from their algal food source and incorporate them as kleptoplasts into their own metabolic
pathways, a phenomenon known as kleptoplastidy. One species showing this ability is *Elphidium excavatum*, a common
foraminifer in the Kiel fjord, Germany. The Kiel fjord is fed by several rivers and thus forms a habitat with strongly
fluctuating salinity. Here, we tested the effects of food source, salinity and light regime on the food uptake (via $^{15}$N and
$^{13}$C algal uptake) in this kleptoplast-bearing foraminifer. In our study *E. excavatum* was cultured in the lab at three
salinity levels (15, 20, 25 PSU) and uptake of C and N from the food source *Dunaliella tertiolecta* (Chlorophyceae) and
Leyanella arenaria (Bacillariophyceae) were measured over time (after 3, 5, 7 days). The species was very well adapted
to the current salinity of the sampling region, as both, algal N and C uptake was highest at 20 PSU. It seems that *E.*
*excavatum* coped better with lower than with higher salinities. The amount of absorbed C from the green algae *D.*
*tertiolecta* showed a marginal significant effect of salinity, peaking at 20 PSU. Nitrogen uptake was also highest at 20
PSU and steadily increased with time. In contrast, C uptake from the diatom *L. arenaria* was highest at 15 PSU and
decreased at higher salinities. We found no overall significant differences in C and N uptake from green algae versus
diatoms. Furthermore, the food uptake at a light/dark rhythm of 16:8 h was compared to continuous darkness. Darkness
had a negative influence on algal C and N uptake, and this effect increased with incubation time. Starving experiments
showed a stimulation of food uptake after 7 days. In summary, it can be concluded that *E. excavatum* copes well with
changes of salinity to a lower level. For changes in light regime, we showed that light reduction caused a decrease of C
and N uptake by *E. excavatum*.

**1. Introduction**
1.1. General information
Foraminifera are unicellular, highly diverse marine organisms known since the early Cambrian (e.g., Scott et al., 2003;
Pawlowski et al., 2003). As major consumers of phytodetritus they play an important role in organic matter recycling in
marine environments, particularly in marine sediments (benthos), from coasts to the deep sea, and in brackish water



(Boltovskoy and Wright, 1976). Most foraminifera are heterotrophic, but some can isolate functional chloroplasts from
their algal food sources, keep them viable in their cells and incorporate them into their own cellular metabolism, a process
termed kleptoplastidy (Bernhard & Bowser, 1999). *Elphidium*, a benthic foraminifera, is one of currently nine known
genera of foraminifera (*Bulimina*, *Elphidium*, *Haynesina*, *Nonion*, *Nonionella*, *Nonionellina*, *Reophax*, *Stainforthia* and
*Virgulinella*) which perform kleptoplastidy (Lopez, 1979; Lee et al., 1988; Cedhagen, 1991; Bernhard and Bowser, 1999;
Correia and Lee, 2000; Grzymski et al., 2002; Goldstein et al., 2004; Pillet et al., 2011; Lechliter, 2014; Tsuchiya et al.,
2015). *Elphidium* has a worldwide distribution and occurs from tropical to Arctic waters (Murray, 1991). This genus
makes up a particularly high proportion of the total foraminiferal population in the shallow water of the Mediterranean,
the English Channel, the North Sea and the Baltic Sea (Murray, 1991). More than 60 morphospecies of *Elphidium* are
known (Murray, 1991), many of which are present in the North and Baltic Seas. The most common species are *E.*
*albiumbilicatum*, *E. excavatum clavatum*, *E. excavatum excavatum*, *E. gerthi*, *E. guntheri*, *E. incertum* or *E. williamsoni*
(Weiss, 1954; Terquem, 1876; Williamson, 1858; Lutze, 1965; Frenzel et al., 2005; Nikulina et al., 2008; Polovodova and
Schönfeld, 2008). *Elphidium excavatum* shows large intraspecific variability (Miller et al., 1982). Two subspecies of this
foraminifer (*E. e. excavatum* and *E. e. clavatum*) have been found to coexist in the Baltic Sea (Lutze, 1965). Schweizer
et al. (2010) showed that these species exhibit large genetic differences with respect to each other and therefore can be
regarded as subspecies rather than as ecophenotypes.
Under anoxic conditions or during longer periods of starvation, kleptoplasts may possibly serve as nutritional source that
can be digested (Falkowski and Raven, 2007). But they can also supplement the nutrition through photosynthesis under
light conditions. Diatoms are the major chloroplast sources for *Elphidium*, with an average of $3.7 \times 10^4$ chloroplasts
possessed by one foraminiferal individuum (Correia and Lee, 2000). The retention time of functional chloroplasts in
foraminifera may vary from several days to several months (Lopez, 1979; Lee et al., 1988; Correia and Lee, 2002).
Experiments with the closely related genus *Haynesina* (Pillet at al., 2011) revealed that these foraminifera can sustain
their kleptoplasts efficiently for more than a week (Thierry et al., 2016). The uptake of kleptoplasts by *Haynesina*
*germanica* through the consumption of diatoms can be seen in the comparison of spectral signatures (Thierry et al., 2016).
Further experiments showed that not all algae are excellent chloroplast donors (Lee and Lee, 1989; Correia and Lee,
2001). It was observed that *Elphidium* absorbs up to five times more chloroplasts from diatoms than from green algae
(Correia and Lee, 2000). It was also pointed out that different light/dark regimes had no influence on the uptake of
chloroplasts by *Elphidium* (Correia and Lee, 2000). Foraminifera below the photic zone can also perform kleptoplastidy
(Bernhard and Bowser, 1999). These aspects show that foraminifera can not only incorporate chloroplasts for
photosynthetic activity, but also benefit from other catabolic mechanisms (LeKieffre et al., 2018).
Currently little is known about the feeding behavior and the C and N metabolism of foraminifera species exhibiting
kleptoplastidy, such as *Elphidium* or *Haynesina*. Moreover, given that plastids may either supplement the nutrition of
foraminifera by providing photosynthates or by being digested, kleptoplastid species may show a slower detrimental
response to starvation, or a slower uptake of (pulses of) algal food (Lintner et al., 2020). Foraminiferal food uptake
depends on several factors such as size of food (Murray, 1963), the type of food (e.g., Lee and Müller, 1973; Nomaki et
al., 2014), the age of the foraminifera and food quality (Lee et al., 1966), water temperature (Wukovits et al., 2017; Heinz
et al. 2012) or salinity (Lintner et al., 2020; Dissard et al., 2009). Salinity and light conditions are highly variable in
intertidal and brackish milieus where foraminifera thrive in highly diverse and active communities. Very little is known
on such light-dark and salinity effects on the feeding behavior of kleptoplastid foraminifera. For example, the kleptoplastid
species *Haynesina germanica* showed no response to changes in salinity while food uptake by the non-kleptoplastid





species *Ammonia tepida* increased with salinity (Lintner et al., 2020). In the same study, both species showed large
differences in the retention of C relative to N, with subsequent adverse effects on the re-cycling of these elements by
mineralization/respiration and excretion to the environment. Such differences, given that these species are (co)dominant
in their foraminifera community, can have important implications on local marine biogeochemical cycles of C and N.
Based on the above mentioned aspects, this study investigated the food uptake and food preference (green algae versus
diatoms) of *Elphidium excavatum ssp.* at different salinity levels and a changing light/dark rhythm. *Elphidium excavatum*
is optimally suited for this purpose, as it is representative for foraminifera in coastal regions and can account for over
90% of the total foraminiferal population in some areas (Schönfeld and Numberger, 2007).
1.2. Sampling location Kiel Fjord
Foraminifera studied here were collected in the Kiel Fjord in northern Germany. The Kiel Fjord covers 9.5 km in length.
It is about 250 m wide in the south (inner Fjord) and widens to the northern part to a width of 7.5 km (outer Fjord) (Nikula
et al., 2007; Polovodova and Schönfeld, 2008). The inner Fjord is about 10 – 12 m deep, whereas the outer Fjord has
more than 20 m water depth. The water in the inner Fjord is well homogenized and has a relatively constant temperature
and salinity at any depth (Schwarzer and Themann, 2003). During the summer months stratification of water masses
occurs, with the surface water having a temperature of 16 °C and a salinity of 14 PSU (1 PSU – practical salinity unit =
1 g salt per liter of water) and the bottom water with 12 °C and 21 PSU (Nikula et al., 2007; Polovodova and Schönfeld,
2008). In the southeast of the Fjord, a fresh water supply, the Schwentine, contributes to a lower salinity of water in this
area. Earlier investigations showed that occasional sea water inflow from the Baltic Sea (very saline surface water with
33 PSU) has no major impact on the hydrography in the Kiel Fjord (Fennel, 1996). The most common sediments in the
fjord are fine sand and dark, organic rich mud (especially found in the inner Fjord). In this area corrosion (abrasion and
redeposition) of foraminiferal tests plays an important role, due to the undersaturation of carbonate in the surface water
(Grobe and Fütterer, 1981).
Over the last 70 years, the Kiel Fjord has been strongly influenced by anthropogenic activities, such as shipyards, military
or infrastructure (Nikula et al., 2007; Polovodova and Schönfeld, 2008). Examples of environmental impacts include high
Cu or Zn values in fish and mollusks (Senocak, 1995; ter Jung, 1992). Furthermore, the Kiel Fjord is rich in nutrients and
organic C. This accumulation of nutrients originates from the city or the surrounding industrial areas and causes a strong
eutrophication in the inner Fjord (Gerlach, 1984). The high input of nutrients leads to a high primary production which,
coupled with the stable water stratification, in turn causes oxygen deficits in bottom water regions (Gerlach, 1990).

**2. Materials**
2.1. Sample collection and culturing
The samples were collected from the Kiel Fjord in northern Germany on 26th and 27th September 2018 with a box corer
on the research vessel F. S. ALKOR. Detailed data on sampling sites are given in Table 1. On board of the research vessel,
the upper 5 – 7 cm of the box corer sediments were wet-sieved through a 63 or 125 µm sieve and kept in storage containers
with seawater from the sampling site until arrival at the laboratory at the University of Vienna (29th September 2018). The
permanent cultures (glass tubes covered with thin foil against evaporation) were kept at constant 20 °C (room temperature)
and at a salinity of 20 PSU in the laboratory.




Tab.1: Information of the sampling points: 1: Strander Bucht, 2: Laboe.

| Sample | N | E | depth [m] | T [°C] | Salinity [PSU] |
|---|---|---|---|---|---|
| **Strander Bucht** | 54°25.998' | 010°11.105' | 16.3 | 14.8 | 20.9 |
| **Laboe** | 54°25.235' | 010°12.409' | 15.3 | 14.9 | 20.9 |


2.2. Preparation of labeled food source
Feeding experiments were performed with the green alga *Dunaliella tertiolecta* and the benthic diatom *Leyanella arenaria*
as food sources. A f/2 nutrient medium (Guillard & Ryther, 1962; Guillard, 1975), enriched with the isotopes $^{13}$C and $^{15}$N
by amendment to a level of 1.5 mmol L$^{-1}$ NaH$^{13}$CO$_3$ and 0.44 mmol L$^{-1}$ Na$^{15}$NO$_3$, was prepared for both cultures. The
algal cultures were kept at 20 °C and a light/dark rhythm of 16:8h in isotopically enriched medium. *Dunaliella tertiolecta*
was harvested at peak biomass, when the cultures showed a strong green color. *Leyanella arenaria* was harvested as soon
as the bottom of the mixing vessel was densely populated and homogenously brown colored. These two states reflect the
characteristics of an optimal culture, where the algae are consumed later preferentially by foraminifera (Lee et al,. 1966).
To collect isotopically enriched algae, the cultures were centrifuged at 800 xg for 10 min. The resultant algal pellet was
washed three times with ASW (artificial seawater, Enge et al., 2011) and centrifuged after each washing step. Afterwards,
the algal pellet was shock frozen in liquid nitrogen and lyophilized for 3 days at 0.180 mbar. In order to retain a high
quality of food, the dried algae were stored in a dry and dark place until use. The labeled algal powder was isotopically
enriched by about 3.3 at%$^{13}$C and 32.3 at%$^{15}$N for *D. tertiolecta* and about 12.6 at%$^{13}$C and 17.9 at%$^{15}$N for *L. arenaria*.
The C:N ratios based on C and N content of the diatom and the green algal food source were 9.14 for *L. arenaria* and
5.78 for *D. tertiolecta*, respectively.
2.3. Feeding experiments
Before the start of the experiments all glassware was cleaned in a muffle furnace (500 °C for 5 h). The "picking tools" and
tin capsules were cleaned with a 1:1 (v:v) mixture of dichloromethane (CH$_2$Cl$_2$) and methanol (CH$_3$OH).
20 foraminifera specimens were collected from the permanent cultures using small brushes and placed in a crystallization
dish with 280 ml sterile filtered sea water from the sampling site in triplicates for the different time points and experiments.
Triplicates were analyzed for each time point and parameter (time, salinity, food source or light condition):
(i)    Salinity: To test the influence of salinity and time on food uptake, the original sea water (20 PSU) was

adjusted by adding NaCl or distilled water to obtain the desired salt concentrations (15, 20 and 25 PSU).

These salinities correspond to different areas of the Kiel Fjord (ca. 20 PSU at sampling location).

Subsequently, foraminifera were incubated for 24 h at 20 °C and a 18:6 h light:dark cycle without food

addition to acclimate to the new parameters, before labelled *D. tertiolecta* food was added. Food uptake was

measured after 3, 5 and 7 days.





(ii)      Food preference: The second experiment investigated the effect of different algal food sources on food
uptake of the foraminifera species. For this, the green algae *D. tertiolecta* and the diatom *L. arenaria* were
offered to foraminifera at 15, 20 and 25 PSU and a light/dark rhythm of 16:8 h and cells collected after 5 d.

(iii)      Light: The third experiment tested the effect of different light conditions on food uptake (only *D. tertiolecta*
food). Here, foraminifera were acclimatized 24 h before food addition to continuous darkness or a 18:6 h
light:dark cycle, at 20 °C and 20 PSU, and samples were collected after 1, 3, 5 and 7 days.

(iv)      Starvation: In order to determine the starvation effect on food uptake of this species, foraminifera were
cultured in the dark without nutritional supplement for different periods of time (1 – 7 days), at 20 °C and
20 PSU, and then were fed for 24 hours with *D. tertiolecta*.

At the end of the test period, foraminifera were picked from the crystallization dishes and any food residues were removed
from the tests. Afterwards, they were washed three times with distilled water. For isotope analysis, foraminifera were
transferred into clean tin capsules (Sn 99.9, IVA Analysentechnik GmbH & Co. KG) and dried for three days at room
temperature. Finally, 5 µl of 4% HCl was added twice to dissolve carbonate from foraminiferal tests. The dissolution was
carried out at 60 °C in a drying oven. Before weighing and isotope analysis, the tin capsules were dried again at 60 °C for
24 h to remove any residual moisture. The dried and weighed samples were stored in a desiccator until isotope
measurements.
2.4. Isotope analysis
Isotope analysis was performed at the Stable Isotope Laboratory for Environmental Research (SILVER) at the University
of Vienna. Ratios of $^{13}C/^{12}C$ and $^{15}N/^{14}N$ were recorded by isotope ratio mass spectrometry (IRMS), using an elemental
analyzer (EA 1110, CE Instruments) coupled with an interface (ConFlo III, Thermo Scientific) to a Delta$^{PLUS}$ IRMS
(Thermo Scientific).
In order to determine the amount of absorbed C or N the at% was calculated according to:

$$\text{at. }\% = \frac{100 \times R_{\text{standard}} \times (\frac{\delta X_{\text{sample}}}{1000} + 1)}{1 + R_{\text{standard}} \times (\frac{\delta X_{\text{sample}}}{1000} + 1)}.$$

(1)

where X stands for C or N here, $R_{\text{Standard}}$: Vienna PeeDee Belemnite $R_{\text{VPDB}} = 0.0112372$ for C, and atmospheric nitrogen
$R_{\text{atmN}} = 0.0036765$ for N.
Since the heavy stable isotopes used as a tracer ($^{13}C$ and $^{15}N$) are also occurring naturally, the natural abundance of these
isotopes needs to be accounted for which was measured in foraminifera that did not obtain labelled algal food sources. To
take this into account, the so-called isotope excess (E) is calculated (Middelburg et al., 2000):

$$E = \frac{\text{atom} X_{\text{sample}} - \text{atom} X_{\text{background}}}{100}.$$

(2)

As $X_{\text{background}}$ isotope abundances of foraminifera were used, which were not fed and thus reflect the natural isotope
abundance signal.
The absorbed amount of isotopes can now be quantified, i.e. labeled $I_{\text{iso}}$ for incorporated C or N.



$I_{\mathrm{iso}}\,\mu\mathrm{g\,mg}^{-1}$ or $\mu\mathrm{g\,ind}^{-1} = E \times C(N)\,\mu\mathrm{g\,mg}^{-1}$        (3)
Here, either the number of individuals (ind$^{-1}$) or the mass (dry matter without test, see 3.1.) of foraminifera were used as
reference.
Finally, we need to consider the different isotopic enrichment of the algal food sources. Thus, "phytodetrital carbon (pC)
or nitrogen (pN)" is calculated accounting for the isotopic enrichment of the food sources. These values are calculated as
follows:
$$pX = \frac{I_{\mathrm{iso}}}{\frac{\mathrm{at.}\,\% X_{\mathrm{phyto}}}{100}}.$$        (4)
2.5. Statistics
To test the main effects of salinity, food source, time, dark: light cycles and starvation, as well as their interaction, on pC
and pN uptake we applied two-way and three-way analysis of variance (ANOVA, 95,0 % confidence intervals). Data
were log transformed when they did not meet normality or homoscedasticity. If the data were significant a Fisher´s LSD
post hoc test was used for more detailed analysis. All statistical tests were performed using Statgraphics Centurion XVI.
The points in the graphs are the mean values from triplicates, with an 2σ error bar for the standard deviation.

## 3. Results

3.1. Effect of salinity and time on C and N uptake from green algal food
The uptake of C (pC) and N (pN) from green algal food sources by *E. excavatum* was slightly affected by salinity (Fig.
1). The statistical evaluation (two-way ANOVA, Tab. 2) showed a marginal significant effect of salinity on pC (log
transformed data; p=0.080), but no significant effect of time (p=0.433) and no salinity x time interaction (p=0.600). pC
tended to be highest at 20 PSU, followed by 25 and 15 PSU across the whole time series. Considering the mean values
after 3 days of feeding, *E. excavatum* showed the lowest pC values at salinities 15 and 25 PSU. The uptake of C showed
a different pattern after 5 days and here reached a maximum at 20 PSU while the values at 15 and 25 PSU were lower but
similar. After 7 days the amount of incorporated C was approximately the same at all three salinities (15, 20 and 25 PSU).
The amount of absorbed nitrogen (pN) was highly significantly affected by salinity (p<0.001) though not by time
(p=0.452). However, the salinity effect interacted significantly with time (p=0.001) indicating that the time kinetics of pN
were different at different salinities. At 15 and 20 PSU N uptake increased steadily from 3 to 7 days while at 25 PSU C
uptake remained constant between 3 and 5 days and thereafter decreased. The values of pN were very similar after 3 days.
This changed after 5 days, where the highest amount of pN was determined at 20 PSU while N uptake was approximately
the same at 15 and 25 PSU (p<0.1). The pattern of pN at this time point (5 days) is highly comparable with the C uptake
pattern. With increasing incubation time the pN values differed significantly. After 7 days (p<0.01), the maximum of pN
was observed at 20 PSU and decreased at 15 PSU and further at 25 PSU.





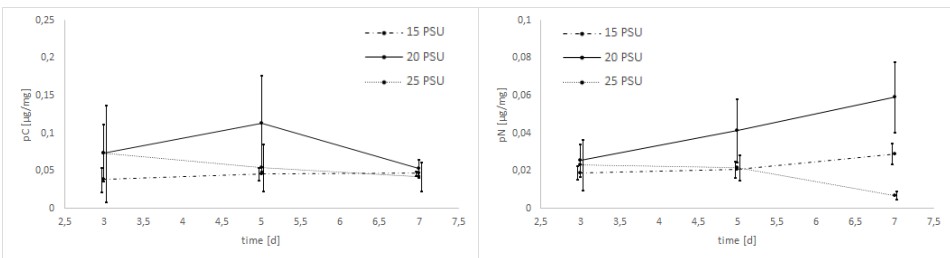


Fig.1: Salinity effects on the uptake of C (pC) and N (pN) from the green algae *D. tertiolecta* by *E. excavatum* after different feeding times at 20 °C and
a light:dark cycle of 18:6 h.

Tab. 2: statistical evaluation of all 4 experiments, significant values are in bold.

| | uptake | Df | Mean Square | F-Ratio | p-Value |
|---|---|---|---|---|---|
| **Experiment I** | | | | | |
| salinity | pC | 2 | 0.667295 | 2.92 | 0.080 |
| time point | pC | 2 | 0.200972 | 0.88 | 0.433 |
| salinity x time | pC | 4 | 0.160959 | 0.70 | 0.600 |
| salinity | pN | 2 | 2.11751 | 19.68 | **<0.001** |
| time point | pN | 2 | 0.0892665 | 0.83 | 0.452 |
| salinity x time | pN | 4 | 0.926354 | 8.61 | **0.001** |
| | | | | | |
| **Experiment II** | | | | | |
| food source | pC | 1 | 0.661141 | 4.00 | 0.069 |
| salinity x food source | pC | 2 | 2.94777 | 17.84 | **<0.001** |
| food source | pN | 1 | 2.31028 | 16.25 | **0.002** |
| salinity x food source | pN | 2 | 0.282117 | 1.98 | 0.181 |
| | | | | | |
| **Experiment III** | | | | | |
| light/dark | pC | 1 | 5.95817 | 16.30 | **0.002** |
| time point | pC | 2 | 0.279452 | 0.76 | 0.487 |
| time x light | pC | 2 | 0.208622 | 0.57 | 0.580 |
| light/dark | pN | 1 | 0.00114997 | 6.43 | **0.026** |
| time point | pN | 2 | 0.000527064 | 2.95 | 0.091 |
| time x light | pN | 2 | 0.00039449 | 2.21 | 0.153 |
| | | | | | |
| **Experiment IV** | | | | | |
| time point | pC | 3 | 0.124304 | 1.65 | 0.158 |
| time point | pN | 3 | 0.142465 | 5.71 | **0.028** |


3.2. Effect of food source (green algae and diatoms) and salinity on C and N uptake
The values of C and N uptake from different food sources at three salinity levels are listed in Table 3.
Tab. 3: The uptake of C (pC) and N (pN) from different food sources (the green algae *D. tertiolecta* and the diatom *L. arenaria*) by *E. excavatum* after
5 days at 20 °C and a light:dark cycle of 18:6 h. The values given correspond to the mean value of triplicates; standard deviations in parenthesis.

| Food source | salinity | pC | pN |
|---|---|---|---|
| *D. tertiolecta* | 15 | 0.0463 (0.0085) | 0.0209 (0.0042) |
| | 20 | 0.1132 (0.0633) | 0.0415 (0.0167) |
| | 25 | 0.0547 (0.0313) | 0.0219 (0.0066) |
| *L. arenaria* | 15 | 0.1231 (0.0647) | 0.0165 (0.0100) |
| | 20 | 0.0877 (0.0206) | 0.0122 (0.0033) |
| | 25 | 0.0780 (0.0330) | 0.0121 (0.0054) |



Two-way ANOVA of log transformed data showed the following: pC tended to be overall higher from *L. arenaria* than
from *D. tertiolecta* sources, indicating some preference for diatom food intake (p=0.069). The salinity effect was highly
significant (p<0.001) and showed a highly significant interaction with food source (p<0.001). Across both food types pC
was lower at 25 PSU than at 15 and 20 PSU. However, this main salinity effect differed by food source: pC from *D.
tertiolecta* peaked at 20 PSU while pC from *L. arenaria* was highest at 15 PSU and showed a sharp decrease at higher
salinities.
Nitrogen uptake showed quite different patterns compared to C uptake. We found a highly significant difference in pN
between food sources (log transformed data, two-way ANOVA; p=0.002), while salinity (p=0.338) and the interaction of
salinity x food type (p=0.181) were non-significant. In contrast to pC, pN was significantly higher after feeding on green
algae than on diatoms. Otherwise, food-specific effects of salinity on pN followed those of pC, i.e. pC peaked at 20 PSU
for *D. tertiolecta* and was highest at 15 PSU for *L. arenaria*.
Comparing the salinity effects on incorporated C and N from feeding with *D. tertiolecta* with those of *L. arenaria*,
different trends can be deduced. The highest pC was reached at the lowest salinity (15 PSU) from the diet with *L. arenaria*
while at higher salinities (20 and 25 PSU) the C uptake was higher when fed with *D. tertiolecta*. In contrast, N was
preferentially incorporated from a diet with *D. tertiolecta*. Such differences in pC and pN from different algal sources
were also reflected in distinct ratios of pC: pN, which were 2.2-2.7 in *D. tertiolecta* and 6.4-7.5 in *L. arenaria*.
3.3. Effects of light regime on the uptake of C and N from green algal food
The experiments clearly showed a strong effect of light regime on the food uptake of *E. excavatum*, with *D. tertiolecta* as
the food source (Fig. 2). Two-way ANOVA of log transformed data showed that the light regime had a highly significant
effect on pC of *E. excavatum* (p=0.002) while time (p=0.487) and the interaction of light x time (p=0.580) were not
significant. Continuous darkness caused a sizable reduction of pC compared to 16:8 h light:dark cycles.
The negative effect of continuous darkness was also observable on pN (p=0.026), and pN tended to increase with time
(p=0.091), particularly so under 16:8 h light:dark cycles. The interaction of light regime x time was, however, not
significant (p=0.153).

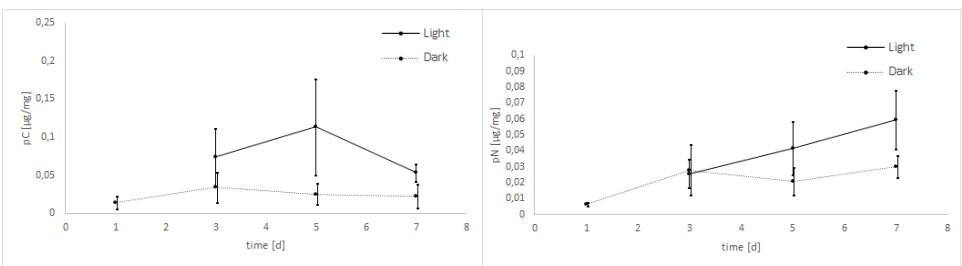

Fig. 2: Effects of light regime (light:dark cycle of 18:6 h versus continuous darkness) on the uptake of C (pC) and N (pN) from the green algae *D.
tertiolecta* by *E. excavatum* after different feeding times at 20 °C and 20 PSU.
3.4. Effects of starvation on the uptake of C and N from green algal food
In a fourth experiment, foraminifera were incubated for different time intervals (1, 3, 5 and 7 days) without any food in
the darkness. After each starvation period they were fed with *D. tertiolecta* and exposed to light for 24 h.



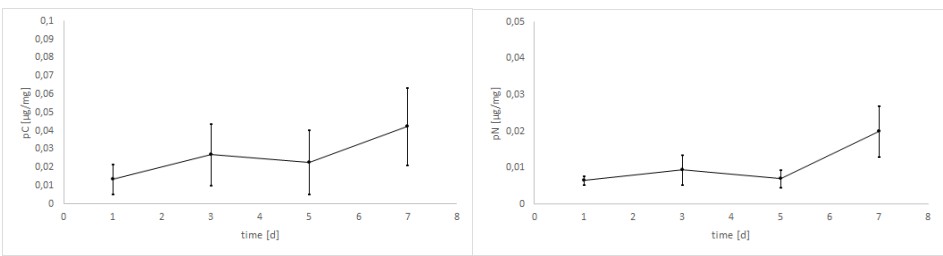


Fig. 3: Uptake of C (pC) and N (pN) from green algal food (*D. tertiolecta*) by *E. excavatum* after different starvation periods in the dark at 20 PSU and 20 °C.

The longer the foraminifera were starved, the more food was consumed within 24 h (Fig. 3). However, one-way ANOVA showed no significant starvation time effect for pC (p=0.158), but there was a significant increase in pN with increasing starvation duration (p=0.028). During the first 5 days in darkness without food, there was hardly any difference in N uptake, while after 7 days in darkness a clear increase of pN was recorded. Similarly, pC tended to be stimulated by prolonged starvation but the variation was too high to become significant.

## 4. Discussion

### 4.1. Food uptake of *E. excavatum* at different salinities

Salinity (15, 20 and 25 PSU) significantly affected the food uptake of *E. excavatum*, especially for longer test times. The low level of ingested *D. tertiolecta* in comparison to other studies with *Ammonia tepida* and *Haynesina germanica* (Lintner et al., 2020; Wukovits et al. 2017) suggests that this green algae was not a preferred dietary source of this foraminifer species. This observation can be compared with experiments by Correia and Lee (2000) which demonstrated an increased absorption of chloroplasts by *E. excavatum,* which corresponds to a dietary preference for diatoms. Though the amount of ingested C from the diatom *L. arenaria* was also low here we found a marginally significant preference of *E. excavatum* for the diatom diet over the green algal diet. It is therefore likely that *E. excavatum* prefers the algal diet that corresponds to the source of its kleptoplasts. Moreover, generally food (C) uptake by a kleptoplastid species (*H. germanica*) was lower than that of a species not showing kleptoplastidy (*A. tepida*) (Lintner et al., 2020; Wukovits et al. 2017), indicating that the chloroplasts can supplement the C nutrition of species exhibiting kleptoplastidy. A shift in food preference in terms of C uptake from diatoms at 15 PSU to green algae at 20-25 PSU is also noteworthy and has not yet been observed in this or other foraminifera species. This might have strong implications on foraminiferal C and N re-cycling in habitats where *E. excavatum* is dominant, given that N retention was approximately 3-fold higher with diets of green algae compared to diatoms (pC:pN was 2.2-2.7 for *D. tertiolecta* compared to 6.4-7.5 for *L. arenaria*).

On a closer look, it can be seen that foraminifera reacted to an increased salt content in the longer term by lower rates of green algal food consumption. The mean C uptake recorded at 20 PSU showed a maximum five days after food addition and declined thereafter. Such a behavior is already known from *H. germanica* (Lintner et al., 2020), a closely related species living in the same habitat. In Lintner et al. (2020) this behavior was explained by the fact that *H. germanica* also contained kleptoplasts, which may serve as internal C and N sources via digestion. In the case of foraminiferal N uptake in our study this effect was not evident, as the amount of incorporated N increased steadily, at least at 15 and 20 PSU. At





this point it should be noted that foraminifera metabolize food C and N during their digestive process and release them
into the surrounding environment as excreta or as respiratory $CO_2$ (Hannah et al., 1994; Nomaki et al., 2014). This needs
to be taken into account the longer an experiment lasts and might explain the decrease in the incorporated amount of C
from day 5 to 7 (Fig. 1). Although C is constantly being absorbed by foraminifera in the form of food, it is also partially
relocated and excreted or released by cellular respiration (Hannah et al., 1994). Furthermore, C can also be used for test
formation. During the preparation of foraminifera for isotope analysis, the test is dissolved in hydrochloric acid and the
amount of incorporated C in the test is not measured, which may cause an underestimation of pC relative to pN at
prolonged feeding times. Although N can also be transferred into various excretions and released into the surrounding
water in organic and inorganic form, a large part still remains in the form of proteins or amino acids in the cell of the
organisms (Nomaki et al., 2014).
After 1 day, foraminifera showed minimum C uptake at the lowest salinity (15 PSU). Comparing the entire time series of
green algal uptake, the 15 PSU series is the only one with a positive slope (k = 0.0021) of pC with time. Based on this
observation, foraminifera might feel uncomfortable at low salinities and react to this with a reduced metabolism. This
may lead to a generally lower activity of foraminifera, which reduces their cell respiration and results in a lower C output.
Foraminifera held at higher salinity (20 or 25 PSU) may have a higher activity and thus a greater C output due to cell
respiration and excretion. The combination of these aspects could explain the negative slopes or peaks of the 20 and 25
PSU trend lines. Direct observations during the experiments showed that foraminifera cultured in crystallization dishes
at 20 or 25 PSU were more mobile (personal observation of crawling observations) than those at 15 PSU. This aspect
confirms the higher activity of foraminifera at higher salinities.
The results of N incorporation differed from those of C. Here, both the 15 and 20 PSU series showed a positive slope with
time while in the long term, less N was absorbed at higher salinities (25 PSU). The magnitude of the slope of the 15 PSU
series was markedly lower than that at 20 PSU. Again, this could be due to the lower activity of foraminifera at 15 PSU
compared to experiments at 20 PSU. However, the decrease of N at 25 PSU with time cannot be explained so easily. A
possible explanation is faster N metabolism coupled to increased excretion of N-containing substances by foraminifera at
high salinity. There are no other studies which are dealing with this arguments, so further experiments are necessary to
resolve this observation. Moreover, the combination of high salinity with an inappropriate diet (green algae) could cause
long-term stress-related damage of the cells. Overall, this experiment highlighted that the digestion and metabolic
pathways of C and N differ substantially and are differentially influenced by environmental parameters in foraminifera
(Lintner et al., 2020; Wukovits et al. 2017).
4.2. Influence of the light/dark rhythm and starvation on the food uptake of *E. excavatum*
Food uptake was affected by light conditions (see fig. 2). Foraminifera had a much lower C and N uptake during
continuous darkness. pC values were low and more or less constant from day 1 through to day 7 (p=0.487). However, N
uptake increased slightly under dark conditions. As already mentioned, *Elphidia* species possess chloroplasts
(kleptoplasts), which they incorporate from their food sources into their own metabolic cycle (Correia and Lee, 2000).
This aspect could be an important contribution to explain the light regime effects on food uptake rates. There are two
different explanations.
First, in complete darkness foraminifera could stop foraging and start feeding on their `own´ chloroplasts. Past
investigations showed that chloroplasts in *Elphidium* were exclusively derived from diatoms, making diatoms their





preferred food source (Pillet et al., 2011). Our experiments showed that *Elphidia* had a significantly higher food uptake
after 7 days of starvation compared to the days before (Fig. 3). During the first 5 days, foraminifera may have either
stagnated with a reduced metabolism or they may have begun to digest their chloroplasts. For further investigations it
would be interesting to detect chlorophyll in foraminifera spectroscopically, since this molecule is found exclusively in
chloro- or kleptoplasts (Cevasco et al., 2015; Krause and Weis, 1991; Mackinney, 1941). One aspect to be discussed here
is the life time of (viable) kleptoplasts in foraminifera under natural conditions. For example, *Nonionella labradorica*
showed a strong seasonal variation in plastid viability (Cedhagen, 1991). According to Cedhagen (1991) specimens of *N.*
*labradorica* collected in February were yellowish and showed no photosynthetic activity. In contrast, individuals sampled
after the spring bloom in March or April were completely green and photosynthetically active. In a study by Cevasco
(2015) foraminifera still contained chlorophyll (>288 photosynthetic plastids) after being held 5 days without food in the
darkness. Lopez (1979) detected functional chloroplasts in *E. excavatum* after 7 days of starvation. The experiments by
Lopez (1979) showed that *E. williamsoni* needs to ingest 65 chloroplasts per hour and individual in order to keep a
constant number of chloroplasts in the cell. It should be noted that the aspect of difference in color mentioned by Cedhagen
(1991) is probably also applicable to our foraminifera. Specimens of *Elphidium* for this study were collected in September,
living in the top few cm of the sediment and showed a yellow coloring. It can therefore be assumed that these individuals
contained fewer functional chloroplasts from the beginning onwards compared to those in the study by Lopez (1979). The
different residence times of kleptoplasts in foraminifera can be fundamentally explained by different feeding and
sequestration strategies as well as diverse digestion abilities (Jauffrais et al., 2018).
Secondly, different food uptake rates under dark or light conditions by *E. excavatum* in this study could be explained by
indirect light effects on chloroplasts in the foraminiferal cells. Since starvation occurred in the dark, no light could
penetrate the tests of the foraminifera and the chloroplasts may therefore have become inactive. However, this raises the
question whether inactive chloroplasts are degraded or stored for some time in order to be able to reactivate them.
Furthermore, it is interesting to know whether *E. excavatum*, which lives in a suboxic milieu like the Kiel fjord, possesses
chloroplasts to acquire oxygen from chloroplast photosynthesis to sustain respiratory metabolism of their mitochondria.
This in turn leads to the question whether *E. excavatum* is viable without chloroplasts or whether the metabolism works
in the long-term only with this additional organelle. To answer these questions clearly further experiments are needed.
According to Jauffrais et al. (2016) the number of chloroplasts in *H. germanica* during starvation periods strongly depends
on illumination conditions. Based on this, foraminifera with kleptoplastidy are more likely to lose active chloroplasts at
light-exposed circumstances (Jauffrais et al., 2016). Combined with the results of Lopez (1979), who stated that
foraminifera must obtain a certain number of chloroplasts from food to maintain a constant number in their cells, our
experiments showed the following: *E. excavatum* is expected to be in a dormant phase under dark conditions, which
entails limited food uptake (Fig. 2). After prolonged starving periods (>7d) in the dark, a starvation effect of this species
is noticeable (Fig. 3). The triggers for this effect are currently unknown. According to Jauffrais et al. (2016) the number
of chloroplasts plays a minor role. It seems that *E. excavatum* can survive in the darkness from the previously ingested
food for up to 5 days of starvation. Only after 7 days of starvation a significantly higher food uptake was observed.
4.3. The influence of salinity and food source on the foraminiferal assemblages in the Kiel fjord
In line with the observations of Lee und Müller (1973) dietary sources used in our experiments had a (marginal) significant
effect on C uptake, with higher C uptake from the diatom food. The effect of food type was even more pronounced for N
uptake, with clearly higher incorporation rates of N from the green algal food (see Tab. 3). However, different salinity



levels caused significant differences with time. Since *E. excavatum* is one of the dominant species in the Kiel fjord
(Schönfeld and Numberger, 2007) and thus plays an important role in the turnover of organic matter, this aspect is
discussed in more detail here.
The Baltic Sea had several transgressive phases that play crucial roles in salinity changes (Robertsson, 1990; Jensen et
al., 1997). The most important salinity indicators in this region are diatoms (Bak et al., 2006; Witkowski, 1994; Abelmann,
1985). Since diatoms serve as the natural food source for *E. excavatum* examined here, their salinity based distribution
plays an essential role in the interpretation of our results. A study by Schönfeld and Numberger (2007) demonstrated the
close connection between foraminifera and diatoms. Their study showed that few days after a phytoplankton bloom of
diatoms a large depositional pulse of organic matter occurred, whereupon the population of *E. excavatum* increased 2 –
6fold. In our experiments we found a slight preference of *E. excavatum* for the tested diatoms (*L. arenaria*) over green
algae (*D. tertiolecta*). Previous experiments showed how certain foraminifera are stimulated particularly by specific food
sources (Lee et al., 1961). However, considering the small amount of incorporated C and N in our experiments, neither
*L. arenaria* nor *D. tertiolecta* belongs to the preferred food sources of *E. excavatum*.
The Baltic Sea is the largest brackish water basin in the world (Voipio, 1981). During the sampling, the salinity was close
to 21 PSU (surface water). This brackish milieu leads to a low diversity of foraminifera (Hermelin, 1987; Murray, 2006).
According to Lutze (1965) benthic foraminifera of this region require a minimum of 11–12 PSU to survive. The lowest
salinity in our experiment was set slightly above this limit, with 15 PSU. Interestingly, the amount of incorporated N was
higher after 7 days at 15 PSU than at 25 PSU, and both pN and pC were highest at 20 PSU (considering mean values of
the uptake). Low salinities or strong salinity fluctuations can lead to smaller test sizes or test abnormalities of foraminifera
(Brodniewicz, 1965; Polovodova and Schönfeld, 2008). Only foraminifera without test abnormalities were taken for
experiments. After the feeding experiments, no visual influence of salinity on test abnormalities or new chambers were
recorded, but the time intervals in this study was likely too short for such observations. The influence of salinity on the
test structure of *Elphidium* in the Baltic Sea has already been investigated (e.g., Binczewska et al., 2018). At our sampling
point in Laboe test abnormalities occur in 12 – 33 individuals per 10 cm$^3$ (Polovodova and Schönfeld, 2008). The authors
suggested a connection between the high number of abnormalities in the Kiel fjord and the salt-rich inflows from the Belt
Sea. The Belt Sea represents the interface where the low-salt Baltic Sea water mixes with the salty Kattegat waters (20-
26 PSU; Hurtig, 1966). At highest salinity (25 PSU) in this study, food uptake apparently decreased over a longer period
of time. Considering the recorded amount of N uptake (Fig. 1) only the 25 PSU series showed a negative correlation and
this trend was neither observed in the 15 PSU nor in the 20 PSU series, which indicates that *E. excavatum* was very good
adapted to the brackish milieu of the Kiel Fjord.
The influences of salinity changes on foraminiferal communities in the Kiel fjord were also investigated by Nikulina et
al. (2008). As discussed before, an increase of salinity probably leads to a decrease of the amount of living *E. excavatum*.
Nowadays, the species *Ammotium cassis* is barely found in the inner Kiel fjord, while a decade ago it was a subdominant
part of the foraminiferal community (Nikulina et al., 2008). This shoes how important changes of the salinity are for
changes in the foraminiferal communities. According to Lutze (1965), *A. cassis* is well adapted to a strong halocline
between the surface and deep waters. Several factors contribute to the formation of a halocline (Steele et al., 1995; Rudels
et al., 1996). Generally, eutrophication and increased storm frequency are important issues in the Baltic Sea (Christiansen
et al., 1996; Seidenkranz, 1993). These factors can lead to a better mixing of the water masses and thus reduce the halocline





and influence the faunal composition. However, the inner Kiel fjord is less saline than the open Kiel Bight and the fauna
is dependent on the salinity of the water (Nikulina et al., 2008; Lutze 1965).
In summary, we found significant differences in food uptake at different salinities. *Elphidium excavatum* seems to cope
better with lower salinities, which correlates very well with the brackish milieu in the Kiel fjord. An increase of the salinity
from 20 to 25 PSU caused more stress for the species than a reduction from 20 to 15 PSU (see reduced uptake of C and
N after 7 days at higher salinities in fig. 1). This once again demonstrates the good adaptation of *E. excavatum* to habitats
of lower salinity. Foraminifera can convert up to 15 % of the total annual flux of particulate organic matter in the Kiel
fjord (Altenbach, 1985). In addition, this region is strongly affected by eutrophication, making the Kiel fjord an interesting
field of research in the future, where interactions of changing environmental parameters with foraminiferal communities
can be studied.

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
