# Peer review of "The effect of salinity, light regime and food source on C and N uptake in a benthic"

_Biogeosciences, 2020_

## Referee Comment (RC1) · Anonymous Referee #1 · 14 Sep 2020

This manuscript describes the effect of different environmental parameters (salinity, light and food source) on the carbon (C) and nitrogen (N) uptake by a benthic klepto-plastic species: Elphidium excavatum. Individuals collected from Kiel fjord were incubated with different type of algae enriched in 13C and 15N, and subsequently analyzed by EA-IRMS to estimate their C and N uptake. Introduction and Material and method sections read well. I have only minor comments for those. However, one of my main concern is the section 4.2 about influence of light, where you seem to assume that kleptoplasts in E. excavatum are photosynthetically functional while this has never been proven. You even cite Lopez (1979) "Lopez (1979) detected functional chloroplasts in E. excavatum after 7 days of starvation.". This is wrong, he did detected chloroplasts

in E. excavatum, but he did not see any C uptake and conclude they are unlikely to be functional and are even merely food items digested by the foraminifera. Please refer to my comment below. If that matter is changed in the text and it is clearly indicated that, yet, there are no proof that chloroplasts from E. excavatum are photosynthetically active, actually it seems from Lopez' (1979) study that chloroplasts in that species are inactive, then I would recommend publication. Despite this concern, I found the study well designed and the manuscript well written. Conclusions are supported by the results presented in the manuscript, and statistical analysis are adapted to the data. Therefore, I recommend publication after some revisions detailed below.

Main comments:

Results

Overall, the result section lacks numbers, i.e. you do not cite numbers to illustrate your description. I would prefer the numbers to appear in the text, especially since you chose to use graphs as figures. We can only infer very approximative values from these. Therefore, I think integrating the exact numbers in the text would ease the reading and the understanding of your data. Also, I found the description sometimes a bit messy and confusing and I recommend rewording some parts (detailed below). Finally, I have a few very minor comments (mainly cosmetics, detailed in the second part of my comments). Line 194: You state that "after 3 days of feeding, E. excavatum showed the lowest pC values at salinities 15 and 25 PSU". While if I look at figure 1, the value corresponding to 25 psu seems higher, even equal to the value of 20 psu. Please clarify. Lines 203 – 204: "After 7 days (p<0.01), the maximum of pN was observed at 20 PSU and decreased at 15 PSU and further at 25 PSU.". I found this sentence a bit confusing, the word "decreased" suggest that the N uptake decreased after 7 days for 15 psu and 25 psu, which is not the case for 15 psu. I think you meant that the value at 15 psu was lower and that the value at 25 psu is even lower. Please clarify. Lines 243-245: Wording is confusing, I suggest: "The negative effect of continuous darkness was also observable on pN (p=0.026). Despite this negative effect, pN tended to increase

with time (p=0.091), particularly so under 16:8 h light:dark cycles." Lines 255 – 259: I found it disturbing that you first state that food uptake increase but that you say the line after that C uptake showed no significance time effect. I think it's misleading in the way that the reader will remember the first key sentence "there is a food uptake increase", while according to your statistical analysis this is not so clear. I agree that from the data it seems that there is a "tendency" of increased C uptake, that could eventually be checked with a longer incubation time (which you could suggest). Please consider rewording this section.

Discussion

The discussion is well written but, in a few occasions, I found that it lacks comparison with existing literature. You keep referring to the same references, papers from your lab, which is fine. But there is other existing literature to be taken onto account. Also, Section 4.2 is very speculative but should remain in the manuscript, but as I mentioned in the introduction, please specify that kleptoplasts from E. excavatum were never proven to be photosynthetically active. Some sentences have thus to tone down. Please find below some suggestions and some references to be added in the text. Lines 287 – 290: Here you mention that C can be transferred to the test and therefore lead to an underestimation of C uptake. I agree that this is important to take into account. Please consider commenting existing publications that recorded a transfer of C from the food source into the calcified test. For example, Lekieffre et al. 2017 (PloS One), measured the assimilation of 13C in the calcite test during a similar experiment involving feeding with 13C-labeled diatoms. Lines 293 – 301: in this section you talk about "reduced metabolism" and stress from different salinity levels. I strongly suggest that you elaborate on that. There are already quite a few examples of reduced metabolism due to stressful conditions in the literature (see Bernhard and Alve, 1996 Marine Micropal, Ross and Hallock, 2016 JFR, etc.). Bernhard, J.M., Alve, E., 1996. Survival, ATP pool, and ultrastructural characterization of benthic foraminifera from Drammensfjord (Norway): response to anoxia. Marine Micropaleontology 28, 5–17. LeKieffre, C., Spangenberg, J.E., Mabilleau, G., Escrig, S., Meibom, A., Geslin, E., 2017. Surviving anoxia in marine sediments: The metabolic response of ubiquitous benthic foraminifera (Ammonia tepida). PLOS ONE 12, e0177604. https://doi.org/10.1371/journal.pone.0177604
Ross, B.J., Hallock, P., 2016. Dormancy in the Foraminifera: A review. Journal of Foraminiferal Research 46, 358–368. https://doi.org/10.2113/gsjfr.46.4.358

Lines 293 – 301: Also, I think it would be very interesting to add a small paragraph about the increase of lipid droplet accumulation observed in benthic foraminifera that encounter stressful conditions. Indeed, you state that "low salinities (. . .) may lead to a generally lower activity of foraminifera, which reduces their cell respiration and results in a lower C output." (line 296). This lower C output could be linked to the accumulation of lipid droplets which seems to be a common response of benthic foraminifera in response to stressful conditions such as anoxia or heavy metals contamination. Here are some references: Frontalini, F., Curzi, D., Cesarini, E., Canonico, B., Giordano, F.M., Matteis, R.D., Bernhard, J.M., Pieretti, N., Gu, B., Eskelsen, J.R., Jubb, A.M., Zhao, L., Pierce, E.M., Gobbi, P., Papa, S., Coccioni, R., 2016. Mercury-pollution induction of intracellular lipid accumulation and lysosomal compartment amplification in the benthic foraminifer Ammonia parkinsoniana. PLOS ONE 11, e0162401. https://doi.org/10.1371/journal.pone.0162401 Frontalini, F., Curzi, D., Giordano, F.M., Bernhard, J.M., Falcieri, E., Coccioni, R., 2015. Effects of lead pollution on Ammonia parkinsoniana (foraminifera): ultrastructural and microanalytical approaches. European Journal of Histochemistry 59, 2460. https://doi.org/10.4081/ejh.2015.2460 Koho, K.A., LeKieffre, C., Nomaki, H., Salonen, I., Geslin, E., Mabilleau, G., Søgaard Jensen, L.H., Reichart, G.-J., 2018. Changes in ultrastructural features of the foraminifera Ammonia spp. in response to anoxic conditions: Field and laboratory observations. Marine Micropaleontology 138, 72–82. https://doi.org/10.1016/j.marmicro.2017.10.011 Le Cadre, V., Debenay, J.-P., 2006. Morphological and cytological responses of Ammonia (foraminifera) to copper contamination: Implication for the use of foraminifera as bioindicators of pollution. Environmental Pollution 143, 304–317. https://doi.org/10.1016/j.envpol.2005.11.033

[Figure]

[Figure]

Section 4.2: Finally, as I mentioned in my general comments, I found disturbing that in some places you seem to assume that kleptoplasts in E. excavatum are photosynthetically functional while this has never been proven. For example, line 341: "chloroplasts may therefore have become inactive". Please state clearly throughout the ms where necessary that kleptoplasts from E. excavatum were never proven to be photosynthetically functional. Therefore, all assumptions based on that are purely speculative. It could also be added that one of the main. Line 331: "Lopez (1979) detected functional chloroplasts in E. excavatum after 7 days of starvation. Âż This statement is wrong. If you read carefully his manuscript, he says twice that chloroplasts are inactive (once in the result section: "The slopes of the lines indicate that Elphidium williamsoni on an average takes up inorganic carbon at a rate about 5 times that in Nonion germanicum, and that the chloroplasts in E. excavatum are inactive with respect to primary production.", and in the discussion: "In accordance with this, no light-induced uptake of 14C-HCO~ by the chloroplasts in E. excavatum could be detected, which means that they are probably merely food items being digested by the foraminiferan.". To my opinion, regarding Lopez' results and your own statement that your specimens were yellowish, your second hypothesis is unlikely. I am fine with you keeping it in your manuscript but Lopez (1979) should be cited correctly and it should be mentioned that E. excavatum chloroplasts were never proven to be functional, and that is rather unlikely.

Minor comments:

Introduction

Line 51: "Under anoxic conditions" is not relevant in this sentence. Lines 57 and 58: I assume "Thierry et al. 2016" refer to "Jauffrais et al. 2016", please make the modifications. Lines 63 – 64: Use conditional here, there is no evidence in the literature yet that this is the case or that kleptoplasts found in foraminifera from below the photic zone are functional. I suggest the following modifications: "These aspects show SUGGEST that foraminifera can not only incorporate chloroplasts for photosynthetic activity, but MAY also benefit from other catabolic mechanisms (LeKieffre et al., 2018)". Lines 97

– 102: I suggest to remove this paragraph as environmental impacts induced by anthropogenic activities is not the subject of your study and is not discussed in the later sections.

Material and methods

Lines 152 – 153: Please specify how many foraminifera were put per tin capsule. Lines 151 – 157: I assume you weighted your capsules before putting the foraminifera inside but this is not mentioned in the text. Please clarify.

Results

Line 199: "while at 25 PSU C 200 uptake (. . .)", I think you meant "while at 25 PSU N 200 uptake (. . .)". Please make the modification. Figure 1: Please remove the half days on the x axis (1.5, 3.5, etc.) as you did for the other figures. All graphs: I would suggest to increase the size of the axis legend. It is yet quite small and it would be nicer for the reader to know immediately which graph is pC or pN.

Discussion Line 262: As you also discussed the effect of the type of food, I suggest to add it in the section title.

Please also note the supplement to this comment:
https://bg.copernicus.org/preprints/bg-2020-306/bg-2020-306-RC1-supplement.pdf

---

## Referee Comment (RC2) · Anonymous Referee #2 · 15 Sep 2020

General comments

The manuscript entitled "The effect of salinity, light regime and food source on C and N uptake in a kleptoplast-bearing foraminifera" by Michael Lintner et al. examined the influence of changing salinity, food sources, light regime, and starvation duration on food uptake by Elphidium excavatum, a kleptoplast-bearing benthic foraminifera. The isotopically labeled food sources were used for the experiment, and C and N uptake were evaluated. This study has fundamental importance on understanding the effects of various environmental factors, especially salinity, on benthic foraminifera, and interpretation of population changes in a highly fluctuating environment like Kiel fjord.

[Figure]

The manuscript is written clearly overall, but the lacks of explanation and careless mistakes are here and there. I feel the experiments are not well designed to investigate some targeted factors. In addition, there found some inappropriate usage of statistics, thus misunderstandings of the results, which makes overinterpretation or inappropriate derivation in the discussion. Unfortunately, I would say this manuscript does not fit to be published in Biogeosciences. The paper would be more improved if the following points are fully considered.

Major points

1. The amount of food

The amount of food provided is not mentioned in the text, so it is hard to evaluate the results. How much was provided? For the time-series experiments (experiment i and iii), was the food enough through the experimental period? If the food availability changes over time, it affects the food uptake accordingly. This point is very important in the experimental design. I assume that the food was provided once (at the start of the experiment). If the experiment is designed to keep the constant food availability, which I think is required in these experiments, please explain how.

2. Food sources

Except for experiment ii (food preference experiment), green algae (D. tertiolecta) was used as a food source despite it is already known that diatom is the more preferred food for the foraminifera (L320–321). Why did you choose green algae as a basic food source in the first place? In addition, in the end, you mentioned that both D. tertiolecta and diatom L. arenaria were not preferable food for E. excavatum (L265, L372). It sounds that the whole experiment was conducted under unsuitable conditions. How should we interpret the food uptake experiment using unfavorable food? I think it's okay that it finally turns out that the foods you chose were not incorporated that much as you have expected (it is how this kind of experiment goes). However, at least you need to explain the reason why these food sources were selected.

**3. Marginal significant effect**

The term "marginal significant effect" is frequently used in the text. The significance level of 0.05 was set as a threshold as you said (L183). Then, the p-values higher than 0.05 should be treated as "not significant". Of course, the threshold is artificial, but that is the way of statistical tests based on the probability of a null hypothesis. You may set the significance level at 0.1, then you can say the p-value of 0.08 is significant, although the significance level of 0.1 is not commonly used in the field of natural science, I believe. In any case, "marginal significant effect" sounds awkward. You can describe the tendency of the results, but if it is not statistically significant, you should say so.

**4. Incorrect result description**

There can be found some wrong description of the results as follows: L224–225: Across both food types pC was lower at 25 PSU than at 15 and 20 PSU. —> This is not correct. According to the results in Table 3, in D. tertiolecta, pC was higher at 25 PSU (0.0547) than at 15 PSU (0.0463). L234–235: The highest pC was...while at higher salinities....the C uptakes was higher when fed with D. tertiolecta. —> Again this sentence is not correct. The data in Table 3 shows that pC in the salinity level 25 was higher in L. arenaria (0.0780) than in D. tertiolecta (0.0547). This kind of error is fatal even if it does not make any difference in overall interpretation or conclusion. Please be very careful when you read the data and describe it. In addition, this kind of description of the data should be accompanied by the numbers (numerical data) and statistical test results. When you say something is higher/lower or increasing/decreasing, please clarify whether it is supported by statistical testing as well.

**5. Data representation**

In the figures, the horizontal axes are all "time [d]", but it seems not appropriate. I suggest to use "Days after food addition" for Figure 1 and 2, and "Starvation duration" for Figure 3. I think it's better to show all three plots (three replicates) in the figures, instead of showing the means and $2\sigma$. Alternatively, please provide all the data used

as supplementary material. For Table 3, I recommend showing the results in a figure (e.g., bar plot with error bars). Numerical information alone is not easy for the readers to read the trends or differences.

Minor points

Usage of PSU: Practical salinity unit (PSU) is not an actual unit of salinity. According to Unesco (1985), the practical salinity scale defined as conductivity ratio has no units. So using PSU as if it is a unit is not appropriate. In addition, L89 "1 practical salinity unit = 1g salt per liter of water" is, strictly speaking, not correct. It is not the definition of practical salinity nor the derivative of its meaning. I think just explaining that you used practical salinity is enough since it is a regularly used representation of salinity in this field. I recommend using the wording like "salinity levels of 15, 20, and 25…" to avoid using PSU.

L18: Leyanella arenaria should be italicized.

L24: light/dark rhythm of 16:8 h —> In the method section, the light regime is 18:6 h. Which is correct?

L47: …shows large intraspecific variability. —> Variability of what?

L54: individuum —> individual

L80: ssp. should not be italicized.

L104: Materials —> Materials and methods

L110–111: How about the light condition? Was in kept in dark or illuminated?

L124: ASW —> It appears once in the text, so it does not need to be abbreviated.

L126–129: Please briefly describe how these values (at% of the food sources and their C:N ratio) were determined.

L133: 20 foraminifera specimens —> How the specimens were selected? Did you use

a certain criterion for specimen selection such as test size, cytoplasm color, cytoplasm filling, activeness, etc... Please clarify.

L136: sea water —> seawater

L137: salt concentrations —> Since it is in practical salinity, salt concentration is not appropriate. How about saying "salinity" or "salinity level".

L138: These salinities correspond to different areas.... ca. 20 PSU at sampling location. —> Please indicate which area corresponds to the salinity levels 15 and 25 as well.

L139: 18:6h light:dark cycle: What kind of light source was used? Did you measure PAR? If so, please include the information. Since the tested species hosts kleptoplasts, information on the light condition is important.

L140: labeled D. tertiolecta food was added. —> How much was provided?

L141: ...measured after 3, 5, and 7 days. —> In the other time-series experiment (experiment iii and iv), food uptake was measured after 1 day as well. Why day 1 was omitted in this experiment?

L150: ...fed for 24 hours with D. tertiolecta. —> Again, how much?

L144: light dark rhythm of 16:8h —> In the caption of Table 3, it says 18:6 h. Which is correct, 16:8 or 18:6 ?

L144: cells collected after 5d. —> Here, do "cells" mean forams? If so, please use "specimens" or "foraminifera" instead.

L152: washed three times with distilled water —> Washing with distilled water may cause the loss of cytoplasm. Was it okay?

L168–172: background value of foraminifera —> How many specimens were used to generate Xbackgroud? Were those foraminifera incubated for 24 h without food (the

time-zero specimens)?

L173–176: ind-1: Individual based values are not used throughout the text, so $\mu$g ind-1 and associating explanation can be removed.

L183: 95,0% —> 95.0%

L184: LSD —> Least Significant Difference

L191: marginal significant effect —> Please do not use this term. See the major comment 3.

L197: highly significantly —> Remove "highly".

L199: At 15 and 20 PSU N uptake increased steadily from 3 to 7 days —> Is it supported by the post-hoc test? If so, please indicate the p-values for both salinity levels.

L199–200: . . .while at 25 PSU C uptake remained. . . —> I assume "C uptake" should be "N uptake", right?

L242, L244: 16:8 light:dark cycle —> In the method and the caption for Figure 2, it says 18:6. Which is correct?

L250–251: It is already explained in the method. To avoid redundancy, this part should be deleted.

L264–266: The low level of ingested D. tertiolecta in comparison to other studies. . . —> If you compare the results of this study to the others, please include the value or range of the food uptake for the referring studies.

L268: marginally significant preference —> If the statistical result is not significant, you cannot say any preference. Therefore, the following sentence, "It is therefore likely that . . .", would not be the inference from the results.

L272–274: A shift in food preference . . . —> I doubt that their result of experiment ii shows any preference of algal types. In the first place, as I pointed out before, pC from

the green algae at salinity level 25 is lower than that from the diatom. So this sentence is not correct from this aspect. In addition, I wonder the discussion of "food preference" from the pC and pN is possible. Since the C and N content of the two food sources differ, the resultant pC and pN may not reflect the preference of food. Moreover, what they detect from their experiment was not the gross C and N incorporated from the diet, but the net C and N in the foraminiferal cytoplasm after respiration and metabolization, as you also mentioned in the discussion. If you really want to know the preference of the diet, different experiments should be designed. Here, maybe the word "preference" is confusing. I think rephrasing the word "preference" to just "uptake" would make sense for some parts. Please consider this point.

L275–276: . . .given that N retention was approximately 3-fold higher with diets of green algae. . .—> The difference in resultant pC:pN (2.2-2.7 for D. tertiolecta and 6.4-7.5 for L. arenaria) may not be accounted for the difference in N retention. Please consider that the original C:N ratio of the food differs for the green algae and diatom. It is shown that the green algae have a higher N (lower C:N ratio) compared to the diatom (L128–129). Therefore, the pC:pN may simply reflect the C:N of each food source.

L293: After 1 day, . . . —> I cannot find the result after "1 day" for the salinity experiment (Fig. 1). In the method, it says "Food uptake was measured after day 3, 5, and 7." So it maybe "After 3 days"? Please confirm it.

L294: the 15 PSU series is. . .with a positive slope. . .—> Is the "positive" slope supported by statistics? According to Table 2, the results from the salinity experiments do not show any significant effect on pC.

L315, 321: Elphidia —> Elphidium?

Section 4.2: I failed to understand the points of discussion in this section. You pointed out the two possibilities to explain the difference in food uptake under different light conditions. One explanation is that under dark condition, foraminifera consumes their own chloroplasts, which results in low food uptake for this group. It is understandable.

As the second explanation (paragraph from L339), you say "indirect light effects", which I assume is about the possibility of the inactivation of chloroplasts. However, how this possibly affects food uptake is not clearly explained. Please clarify this point.

L336: ...these individuals contained fewer functional chloroplasts from the beginning... —> This sentence ruins the earlier discussion in this paragraph. It somehow sounds strange for me that this study focuses on kleptoplast-bearing foraminifera, whereas the specimen used contained fewer plastids (although whether the plastids content is enough cannot be evaluated). I think the discussion relating the function of kleptoplasts should be toned down if you think the specimens used do not represent the general feature of kleptoplastidy.

L357: ..(marginal) significant effect —> It should be treated as "not significant" as I noted before. Please reconsider the tone of the discussion based on such results (results representing some trends but with no statistical support).

L369: ...we found a slight preference ...for the tested diatoms ...over green algae... —> Talking about "preference" is not appropriate. Please see the above comment for L272–274.

L378–389: In this part, the influences of high salinity on test morphology (size and abnormalities) are addressed. However, I failed to understand how it relates to the results of this study (i.e., the effect of salinity on food uptake). I assume that you try to relate the higher stress at higher salinity from the aspect of morphology and food uptake. However, since no test abnormalities was observed at the highest salinity level in this study, I suggest to tone down this part. I think it is also safer to tone down the last sentence, "E. excavatum was very good adapted to the brackish milieu of the Kiel Fjord (L389)".

L393: shoes —> shows

Figure 2: In my understanding, the "light data" is the same as the one in Figure 1

(salinity level 20). Please clarify it in the caption. Moreover, the plots of the "dark data" and their error bars are not aligned vertically. Please correct it.

A schematic diagram illustrating the experiments is very helpful to understand this study. Please consider to add this kind of figure.

I hope my comments above would be helpful.

---

## Referee Comment (RC3) · Anonymous Referee #3 · 18 Sep 2020

Manuscript ID: bg-2020-306 Title : The effect of salinity, light regime and food source on C and N uptake in a kleptoplast-bearing foraminifera MS Authors: Michael Lintner, Bianca Lintner, Wolfgang Wanek, Nina Keul, and Petra Heinz

Global comment: The manuscript by Lintner et al. investigates under controlled conditions the effect of salinity, light regime and food source on C and N uptake in a kleptoplast-bearing foraminifera. This is a relevant topic for all marine biologists working on benthic foraminifera species especially in brackish habitats. It will be also of interest for some biogeochemists working on forams. I found that the paper had interesting data but I had some difficulty to link the ability to deal with variable salinity and

the presence and discussion around the ability of Elphidium excavatum to keep or not functional kleptoplasts.

Major Concerns: Elphidium excavatum is now considered as a species complexes (Darling et al 2016. https://doi.org/10.1016/j.marmicro.2016.09.001). Please could you add SEM or genetic data to ascertain the species: E oceanese or E selseyense or E. clavatum …. See darling et al.. This is highly important if you want to discuss the ability of an Elphidium to use or not its kleptoplasts as there is a large difference between Elphidiidae regarding their ability to use or not their kleptoplasts (discussed in Jauffrais et al 2018. https://doi.org/10.1016/j.marmicro.2017.10.003)

In the discussion: line 331 .Please modify the use of the reference to Lopez et al. 1979, as she demonstrate that I cite Inorganic carbon "Uptake could not be demonstrated in E. excavatum." To my knowledge, there is no article that demonstrate the functionality of kleptoplast in Elphidium excavatum. In the introduction and discussion some recent and relevant articles on Elphidium, kleptoplasty and feeding behavior/strategy of forams are missing: Chronopoulou et al 2019 (https://doi.org/10.3389/fmicb.2019.01169) , Jauffrais et al 2019 (https://doi.org/10.1093/femsec/fiz046), darling et al 2016, Jauffrais et al. 2017 (https://doi.org/10.1371/journal.pone.0172678), tsuchiya et al 2020. (https://doi.org/10.3389/fmars.2020.00585), Salonen et al. 2019 https://doi.org/10.1038/s41598-019-48166-5 I have some concern with the microalgae used to feed the forams, could you argue about this choice and how much algae did you give? The experimental part is not well explained and with the actual information, it is impossible to repeat the experiments (food?, light?...). Other minor comments Introduction: Line 57 & 58: Thierry et al 2016 should be Jauffrais et al 2016. Not only H. germanica but also E. williamsoni. Not only spectral signatures but pigment composition and also DNA for E. williamsoni.

Materials Line 106: could you specify the light penetration depth in the fjord? As mentioned by many authors light history is highly important regarding kleptoplast retention

and functionality. Line 131: Could you specify the time between sampling and the experiments? Line 142: how much food did you give? Every day? Line 145: How much light did you give during the experiment? The source of light? How did you measure it? Line 151-152: the use of distilled water may cause an osmotic shock and the loss of N and C compounds. Did you observed this effect or consider it? How many forams did you used for the isotope analysis?

Results Could you increase the size of the figures. Table 2 exp 2, salinity is not mentioned and tested , why?

Discussion Line 321: you cannot extrapolate your data to all Elphidia. Line 331: please read again Lopez et al, what you wrote is not correct. Line 336. You cannot say that with your data. It seems that you are making a confusion between presence of kleptoplast and functionality. You did not measure the functionality of your kleptoplasts. To my concern, you cannot even say that they contain kleptoplasts. The presence of kleptoplast is highly variable in kleptoplastic species and in some species quickly digested... Line 353. According to Jauffrais et al. (2016) the number of chloroplasts plays a minor role. What do you mean? Line 368. You speak of planktonic diatom, thus when the bloom is over they are dying, so forams are feeding on dead diatoms. How can forams use this kleptoplasts?

―――――――――――――――――――――――――

---

## Author Comment (AC1) · 10 Oct 2020

Thank you for your feedback. It is a very good and important review, which leads – after the adaptation of the manuscript – to a better understanding of the paper.

Generally: We rephrased section 4.2. and adapted the citation of Lopez (1979). Besides, we included the aspect that chloroplasts in E. excavatum may not be functional due to the observation of Lopez (1979).

Results: We added values in the text, for a better description of our results. Additionally, a table of all pN and pC values will be given in the supplementary part. Line 194: The

difference in mean pC between the 20 (0.07318) and the 25 (0.07306) PSU levels is very low, but our statement is correct. It will be more clearly, after we added mean values to the text. Lines 203-204: We corrected these sentences. Lines 243-245: We improved the text here. Lines 255-259: An increase of food uptake considering pC is just seen if you look at a correlation line based on the mean values and not via ANOVA. We added more information to this paragraph and also the aspect of a "tendency" of food uptake.

Discussion: We added all your suggested literature and also rephrased section 4.2. (see above). Lines 287-290: We added this information and reference to the manuscript. Lines 293-301: We added more references, and also expanded the discussion by including the aspect of the accumulation of lipid droplets. Section 4.2.: As described above, we have rephrased this section and inserted all your comments.

Minor comments: Line 51: We skipped this phrase. Lines 57-58: We adapted the literature. Lines 63-64: We modified these sentences. Lines 97-102: We removed this paragraph. Lines 152-153: 20 foraminifera were put per tin capsule – we added this information. Lines 151-157: Yes, we did this – we also added this information to the manuscript. Line 199: Thank you for this hint, indeed we meant N uptake, we adjusted it in the text now. Figure 1: We adapted this figure to your suggestion and also increased the size of the axis legend. Line 262: We added this information to the section title.

---

## Author Comment (AC2) · 10 Oct 2020

Thank you for your feedback. We cannot agree with you, that the experiments are not well designed to investigate targeted factors. Highly significant results were generated concerning the tested factors, which would be not possible if experiments were not well planned. Furthermore, we cannot agree with your statement of using inappropriate statistics. As described below, the usage of the term "marginal significant effect" for a p – value of 0.080 (significance level 0.05) may not correspond to some conventions but p=0.080 is definitely a "trend" and so we can deal with this result. This "trend" allows us to speculate about the C uptake of foraminifera and therefore the accusation of

overinterpretation/misunderstanding of the results seems inappropriate. Additionally, N uptake shows even a highly significant difference (p<0.001) and therefore justify statements like "significant differences of salinity based food uptake". To avoid further misunderstanding, we clarified our statements in the discussion.

Major Points 1. The amount of food: The food (5 mg) was provided at the beginning of the experiment. At the end of the experiment there was enough food remaining in the dishes, otherwise the experiments would make no sense. Therefore, sufficient food was available for foraminifera during the whole experiment.

2. Food source: D. tertiolecta is very commonly and successfully used for feeding experiments in other studies (Lintner et al. 2020, Wukovits et al. 2018, 2017, Grabenstatter et al. 2013, Linshy et al. 2014, Nomaki et al. 2006, Heinz et al. 2002, Lee et al. 1961, . . .). We use this algae for many years in our culture lab. In addition using D. tertiolecta enables us to compare quantitatively our results with previous studies investigating other species or variations with other factors. Please keep in mind, we assumed that both algae are not a preferred food source, due to the low uptake values in comparison to other tested foraminiferal species. We speculated, that this low uptake may be due to the unfavorable food source, but since no one has yet examined the food uptake of E. excavatum we cannot compare our values. Maybe E. excavatum has generally a lower uptake of food than other foraminifera. But we rephrased this in the discussion so it becomes clearer to everybody. Also, we would be very careful to use this pC and pN values as absolute values! We think the activity of foraminifera depends strongly on seasonal fluctuations.

3. Marginal significant effect As described in the introduction, we rephrased this part and changed "marginal significant effect into "trend".

4. Incorrect results description We corrected the mistakes and added the values to the text.

5. Data representation We changed axes titles as suggested by reviewer 2, although

"time (d)" would be more consistent to other studies published in Biogeosciences. For Fig. 3 we can think about a replacement of the name. We think it is very confusing to plot all points. This way you would have 9 data points at a single x-value and this is not easy to read. All data would be provided as a supplementary file in the final published version. We produced a bar plot for the numbers in tab. 3.

Minor points We will check this PSU-problem and will than rephrase the paper. L 18/24/54/80/104/124/136/173/183/184/191/197/199/250/264/293/315/321/369/393: We corrected this. L 47: Morphological variability – we added this to the text. L 110: Illuminated, we added this in the text now. L 126: The calculation of the values is described in detail at 2.4. Isotope analysis L 133: We took foraminifera >150 $\mu$m and only individuals which tests were totally filled with cytoplasma – we added this information to the text. L 137: Also a modified seawater has a salt concentration and therefore it would be ok to use salt concentration, but we adapted the text here. L 138: 15 PSU to the "Schwentinemündung" and 25 PSU to the outer Fjord. The information is now added to the text. L 139: We used fluorescence tubes from the incubator as a light source with 30 $\mu$mol photons m-2 s-1. The information is now added to the text. L 140/150: 5 mg/cristallisation dishes. The information is now added to the text. L 141: We know from other studies (not published by now) that after 1d you have no significant different food uptake by E. excavatum if you change the environmental parameters (light intensity, heavy metal concentration, ...). L 144/242/244: 16:8 is correct and was corrected in the text L144: We replaced the term "cells" by "foraminifera". L 152: It is okay. We did not see any tests broken up due to osmotic shock. L 168 – 172: For background values, we used 20 foraminifera for one data point – also triplicates were done. These foraminifera were taken freshly from the main culture. These foraminifera were not incubated to prevent contamination of this individuals with isotopes! L 199: This is right – we adapted the text here. L 268: We rephrased this. L 272: We considered this point and changed "preference" into "uptake". L 275 We proofed this and adapted this point. L 294: Yes, statistics was added. Section 4.2.: The second part was rephrased, maybe now it becomes more clearly. L 336: This part was also rephrased. But generally, if individuals contain fewer plastids it does not mean that they have no plastids, therefore they can also represent the general feature of kleptoplasts with even fewer plastids. L 357: We treated it as a "trend". But again, it will not change the interpretation of the results here. L 378: We adapted this part to your suggestions. Fig. 2: It was added to the caption. The error bars are deliberately shifted to avoid overlapping with the "light-data". If you look carefully at fig. 1 it is the same and I think it is very useful to have more information about the error bars.

---

## Author Comment (AC3) · 10 Oct 2020

Thank you very much for your input and your ideas. Major Concerns: We added some SEM pictures and clarified this. Discussion: The part with Lopez was rephrased; also we added this suggested literature; we used this algae because we had stable cultures from both algae at our laboratory. Therefore we were able to produce isotopic labelled food sources. Another argument was, that D. tertiolecta is frequently used in other studies and so we were able to compare and discuss these results better. We fed with 5 mg algae per cristallisation dish and used a light intensity of 30 $\mu$mol photons m-2 s-1. We added this information to the text. All minor-comments were considered in the

new manuscript version.

Materials: Line 106: Light penetration depth was added. Line 131: The experiments started 4 days after the sampling of the sediment. Line 142: Food addition was 5 mg per cristallisation dish at the beginning of the experiments. After the experiments sufficient food still remained in the dishes so food was not a limiting factor. Line 145: We used 30 $\mu$mol photonen m-2 s-1 from a fluorenscent tube. Line 151: 20 foraminifera were used for 1 data point. At each combination of time and salinity we produced 3 data points (triplicates). We noticed no breakup of the tests during the washing steps, therefore we could say there was no loss of C and N.

Results: We increased the size and added the information to tab. 2.

Discussion: Line 321/331: Text was adapted. Line 336: We agree that we cannot state this, but that's the reason why we "assumed" it – to avoid misunderstanding, we rephrased this part. Line 353: The number of chloroplasts plays a minor role for food uptake – we changed the text here. Line 368: At this point we were just discussing the food preference of E. excavatum. Of course they are not able to obtain kleptoplasts from dead diatoms. Foraminifera can just use chloroplasts from living benthic diatoms.

---

## Editor Decision (ED1)

Statistics:
In my understanding, the data do not follow the normality or homoscedasticity or both, as you used log-transformed data for ANOVA (for experiments 1, 2, 3). I think it is dangerous to discuss based on means in such cases. In the first place, parametric testing using a very small sample size (n=3) which does not satisfy the prerequisites on distribution should be avoided. Why not using non-parametric testing instead?
In addition, post-hoc multiple comparison by Fisher's LSD should not be used for more than 4 groups (since this method does not consider p-value adjustment). Fishers LSD test has been criticized for not sufficiently controlling for Type I error.
If the non-parametric testing will be conducted, and only if it still shows the same outcome, then I would recommend publication after revision.

p-values interpretation:
In the reply comment, you said "p=0.080 is definitively a "trend" and so we can deal with this result". Sometimes I also find papers using this kind of reasoning, but it is not correct. Please read the following,

ASA's Statement on p-Values: Context, Process, and Purpose. The American Statistician, Vol 70, Issue 2, 2016.
It says,
・"A P-value, or statistical significance, does not measure the size of an effect or the importance of a result"
・"P-values do not measure the probability that the studied hypothesis is true, or the probability that the data were produced by random chance alone".

Statistics are very helpful for describing and interpreting the results and should be used, but only be used properly. I think it is most important to look at your data themselves and on the variance of each group very carefully. The attitude that solely relies on "statistical significance" is dangerous.

Unit of salinity:
permil is not appropriate as well… As I pointed out before, salinity has no unit. PSU, rather than permil, was still better, though it is not needed. Please follow the guideline of the journal for unit representation if any.

Table 3:
In the previous review, I recommended representing the data as a figure. I still think it should be so.

---

## Author Response (AR2)

Response to the reviews:

We have added some additional information about the problem using distilled water. In general, all samples were always treated equal, which means they were all washed with the same volume of distilled water. Therefore, any potential impact that may have arisen from using distilled water had the same effect on all samples. But we worked very carefully (because it is not the first time) and therefore no tests were burst, which means no cytoplasm were extracted or lost.

Line 57: is correct now.

Line 57 to 59: Reference was added.

Line 85: corrected.

Line 167: We have not done any genetic studies. But after a morphological analysis (which is quite common to identify foraminifera) our used foraminifera look quite similar to that S5 type (called *E. selseyense* ) of Darlin et al. 2016. But further in all common literature, this foraminifer is called *E. excavatum* (see references in the manuscript). Therefore, we decided to use *E. excavatum* also in our manuscript. After the last review process, we added a very detailed REM picture which allows further readers to check which foraminifera we are dealing with. This was really important, due to the fact that foraminiferal names change at the moment dramatically fast. So, we added to the manuscript: "After Darling et al. (2016) our tested foraminifera are called *E. selseyense*. Actually *E. selseyense* is officially accepted as *Cribroelphidium selseyense* (WORMS - Heron-Allen & Earland, 1911). But due to the high importance of the "older" name we used for this manuscript the most common and more often cited name *E. excavatum*."

Statistics: We used now the non-parametric test (Kruskal-Wallis) and can show (of course) the same output.

Units of salinity: To avoid further misunderstandings, we modified the text after the suggestions of the reviewer.

Table 3: Table 3 is now Figure 3 and shows the data via box plot. The values are now presented in the appendix.

---

## Editor Decision (ED2)

Manuscript ID: bg-2020-306
Title : The effect of salinity, light regime and food source on C and N uptake in a benthic 1
foraminifera MS Authors: Michael Lintner, Bianca Lintner, Wolfgang Wanek, Nina Keul, and
Petra Heinz

Comment:
The authors replied properly to the majority of my previous comments and improved the
materials and methods by adding much more information.

However, references are not always properly used and I still have some concerns with the
species name that they choose for this Elphidium and with the use of distilled water to wash
the forams before analysis that may cause, to my concern, an osmotic shock and the loss of
N and C compounds (e.g. Movellan et al., 2012. Protein biomass quantification of unbroken
individual foraminifers using nano-spectrophotometry.
Biogeosciences 9, 3613-3623.).

line 57: Thierry et al 2016 should be Jauffrais et al 2016.
Line 57 to 59, If you add E. williamsoni please also add the corresponding reference
(https://doi.org/10.1093/femsec/fiz046).
Line 85 spp.??
Line 167: If the Elphidium is the genetic type S5 after Darling et al, it should be E. Selseyense
and not E. excavatum, which is a species complex. Please clarify this point.